# Development of an Expanded Snack of Rice Starch Enriched with Amaranth by Extrusion Process

**DOI:** 10.3390/molecules24132430

**Published:** 2019-07-02

**Authors:** Lilisbet Castellanos-Gallo, Tomás Galicia-García, Iván Estrada-Moreno, Mónica Mendoza-Duarte, Rubén Márquez-Meléndez, Beatriz Portillo-Arroyo, Cesar Soto-Figueroa, Yarely Leal-Ramos, Daniela Sanchez-Aldana

**Affiliations:** 1Faculty of Chemical Science. University Campus II. Food Science and Technology Programme. Autonomous University of Chihuahua, Chihuahua, Chih. CP. 31125. Mexico; 2Research Center for Advanced Materials. CIMAV-Chihuahua. Miguel de Cervantes 120, Complejo Industrial Chihuahua. Chihuahua, Chih. CP. 31136. Mexico

**Keywords:** rice starch, expanded snack, extrusion process, amaranth

## Abstract

This study aimed to obtain a second-generation snack by extrusion from the by-product of rice milling enriched with amaranth. The raw material used was amaranth flour (AF), rice starch (NS) and modified rice starch (MS), which were evaluated by the analysis of substitution degree (SD), differential scanning calorimetry (DSC), viscosity (RVA), Fourier transform infrared spectroscopy (FT-IR) and X-ray diffraction (XRD). The snacks were expanded by extrusion and microwave oven, as a reference method. The samples were evaluated in hardness (D), expansion index (EI), apparent density (DAP), and protein content (P). Afterward, the optimized samples were evaluated by scanning electron microscopy (SEM) and resistant starch (RS). During the thermal characterization, a clear trend in the decrement in gelatinization temperatures was observed (78.35 to 63.90 °C in NS and MS respectively). The curves obtained in RVA analyses showed typical behavior of native (6.35 Pa.s) and extruded starches (2.88 Pa.s), with a significant decrease in viscosity peak. Through the analysis of FT-IR, the introduction of the functional acetyl group (stretching at a wavelength of 1735 cm^−1^) was corroborated. Snack samples results showed a maximum hardness in MS, with a value of 121 N, and the NS (100%) presented the highest EI value (1.41). The lowest DAP values were obtained for the MS (0.48 g/cm^3^, 100%) and AF (0.49 g/cm^3^, 100%) samples. P increased to a higher concentration of AF. In the optimum formulation, the SEM image showed that the expanded microwave sample increased the porosity and obtained an RS value of 8.2%. The formulation obtained in the present study presents high characteristics to be used in the development of a healthy snack.

## 1. Introduction

Obesity and overweight are currently a challenge for public health in the world. This condition is associated with the development of chronic diseases such as diabetes mellitus type 2, hypertension, cerebrovascular diseases, and cancer. Its increase is directly linked to the consumption of foods with high caloric density and nutrient deficit, as well as little or no physical activity [1,2]. Urbanization and the rise in income have been a strong influence on the population to substitute traditional meals, which are rich in complex carbohydrates, for diets with simple sugars, fats and products of animal origin. An increase in the consumption of ready-to-eat foods, soda drinks and meats are triggers in the development of obesity and overweight. Although there is no evidence to relate it to the regular intake of snacks, it is said that the selection of snacks with high empty calorie content promotes it [3]. The food industry is focused on the development of snacks formulated with pseudocereals since they are alternative raw materials that contain high-quality proteins, fiber and bioactive compounds. The biggest drawback in the production of these snacks is obtaining texture properties similar to those already established in the markets. About this, the snacks modification and addition of nutrients utilizing conventional processes of production and at the same time, the achievement of attractive sensory properties is a great challenge to overcome. One of the alternatives for snacks production is the thermoplastic extrusion. It is a process that allows the mixing of different raw materials favoring the obtaining of enriched products and the retention of nutrients due to the combination of high temperatures and short time, causing changes in the structure of starches and proteins mainly [4,5]. On the other hand, those products rich in starch are the main raw materials in the preparation of snacks; however, native starches have a low resistance to shear, thermal decomposition, high retrogradation, and syneresis [6], limitations that reduce their use at the industrial level. The chemical modification by acetylation is an alternative to solve these problems. Acetylation is an esterification of hydroxyl groups in the anhydroglucose unit of the starch molecule. It is used in order to improve the physical, chemical and functional properties of native starch as water absorption, gelatinization and retrogradation. During the acetylation process, the hydroxyl groups (OH-) of the glucose monomers are transformed into acetyl groups (CH3COO-). The extrusion process is an alternative method for realizing this modification of starch where there are advantages such a less use of reagent, no polluting effluent production, and short process time, compared with the traditional method. There is also the possible formation of resistant starch during extrusion due to the high temperatures and shear forces that lead to the destruction of the structure and subsequent realignment of the chains of polymers mainly amylose, this starch is known as retrograde starch or resistant starch type III [7]. Its combination with chemically-modified starch gives rise to a resistant starch with superior stability and reduced digestibility favoring the functioning of the colon [8]. The purpose of the present investigation was to evaluate a directly expanded snack obtained by extrusion using as raw material native starch extracted from broken rice, modified rice starch and amaranth flour. The mixture design was employed for the optimal physical properties of a snack with high potential for the development of healthy snacks with benefits for the consumer.

## 2. Experiment

### 2.1. Materials

Cracked and broken rice *(Oryza sativa),* a subproduct of milling rice, was used, as well as puffing amaranth *(Amaranthus spp)*, acquired in the local market (Chihuahua, Mexico).

### 2.2. Obtention of Rice Flour and Amaranth

The amaranth and rice flours were obtained through a dry grind in a rotor mill (Retsch, SK 100, Haan, Germany) and subsequently sieved in a 60 mesh (250 μm). The yield was around 90%.

### 2.3. Isolation of Rice Starch

Starch isolation was performed based on the methodology described by Santos et al. [9], where a solution of sodium hydroxide (0.25%) (MACRON, CDMX, Mexico) was added to the flour at 25 °C at a ratio of 1:6 (*w*/*v*), and it was kept stirring for 1 h, then it was neutralized with a solution of hydrochloric acid (HCl, 10%, *v*/*v*). The mixture was left to settle for 24 h cooling and after the time the supernatant was removed and the sediment was centrifuged (Centra CL3R, Thermo IEC. EUA) at 4000 rpm (2750× *g*) for 10 min, making three consecutive washes with distilled water. The sedimented starch was dried in an oven (Ecoshel FCD 3000, Mc Allen, TX, USA) at 40 °C for 12 h. Finally, the sample was milled in a rotor mill (Retsch GmbH, Haan, Germany) to obtain a particle size of 250 μm (60 mesh).

### 2.4. Acetylation of Rice Starch

This modification was made based on the methodology described by Mali and Grossmann [10]. Anhydride acetic (JTBaker^®^, CDMX, Mexico) was used as an esterifying agent, where 500 g of native starch was adjusted to a humidity of 22% and then 5.5 g of acetic anhydride was added. The sample was stored for 12 h. After that time, 50 mL of NaOH solution was added at 10% (*w*/*v*) until the pH was adjusted in a range of 8.5 to 9. The acetylated sample was processed in an extrusion machine (C.W. Brabender Instruments, Inc., Duisburg, Germany) of a single screw with a 1:1 compression ratio, screw speed of 50 rpm and a die diameter of 9.5 mm. The temperatures used were 60, 70, 80, and 100 °C in zones 1, 2, 3, and 4 respectively. The extrudate was dried in an oven (Ecoshel FCD 3000, Mc allen, TX, USA) with air circulation at 40 °C up to a humidity of 8%. Finally, it was ground in a rotor mill (Retsch SK 100, Haan, Germany) and sieved at 60 mesh (250 μm).

#### Degree of Substitution (DS)

It was carried out based on the methodology described by Bello-Pérez et al. [11]. One gram of modified starch was placed in a 250 mL flask and 50 mL of an ethanol-water solution (75% *v*/*v*) was added. The mixture was kept under stirring for 30 min at a temperature of 50 °C. Once the cold mixture was added, 40 mL of potassium hydroxide (0.5 N) (MACRON, CDMX, Mexico) was added and it was kept at rest for 72 h, with occasional agitation. After the time, it was titrated with a standard solution of HCl (0.5 N), using phenolphthalein as an indicator. Simultaneously, a control sample was titrated using native starch. By means of the following Equations (1) and (2), the percentage of acetylation and the degree of substitution (DS) were obtained:(1)$$\%\;Acetylation=\frac{\left(\left[mlcontrol-mlsample\right]*0.5*0.043*100\right)}{\mathrm{weight}\;\mathrm{of}\;\mathrm{the}\;\mathrm{sample}\;\mathrm{on}\;\mathrm{dry}\;\mathrm{basis}}$$
(2)$$DS=\frac{162*\%\;Acetylation}{4300-\left(42*\%\;Acetilation\right)}$$

This analysis was performed in triplicate in each of the treatments.

### 2.5. Differential Scanning Calorimetry (DSC)

The thermal characterization was carried out according to the methodology described by Paredes-López et al. [12] using a calorimeter (NETZSCH DSC 200 PC Phox^®^, Selb, Germany). Two milligrams of the sample were weighed directly into 40 μL aluminum trays, and 20 μL of deionized water was added with a micropipette to obtain a solid suspension with a water content of 65–75% (*w*/*w*). The trays were sealed through a press (NETZSCH DSC) and allowed to equilibrate for 1 h at room temperature. The analysis was carried out at a heating rate of 10 °C/min, and a temperature range of 30 to 140 °C, with a sensitivity of 0.005 mcal/s. An empty tray was used as a reference for all measurements, the equipment with indium being previously calibrated. The transition temperature (initial temperature T_0_, gelatinization temperature T_g_ and final temperature T_f_) and enthalpy of gelatinization (ΔH_g_) were determined using NETZSCH Proteus^®^ Analysis software. This analysis was performed in triplicate in each of the treatments.

### 2.6. Pasting Properties (RVA)

Pasting properties were determined based on method 61-02.01 [13]. The analysis was performed on a rheometer (Anton Paar Physica MCR 501, Graz, Austria). The maximum viscosity (μ_max_), minimum viscosity (μ_min_), final viscosity (μ_f_), and the retrogradation viscosity (μ_r_) were determined to evaluate the rheological properties of the structure of the flours and starches. For the measurement, 40 mL of distilled water and 8.69 g of starch were weighed. The temperature profile used for the analysis consisted of a heating phase of 25 to 90 °C, at a speed of 2 °C/min, followed by a stability phase in which the sample was maintained at 90 °C for a time of 10 min to finish with a cooling stage of 90 °C to 25 °C at 2 °C/min; the shear rate was kept constant at a value of 210 s^−1^ throughout the analysis. Rheoplus Software was used for the analysis of the results. This analysis was performed in duplicate in each of the treatments.

### 2.7. Infrared Transmission Spectroscopy with Fourier Transform (FT-IR)

The characterization of native and modified starch was carried out according to the methodology reported by Andreuccetti et al. [14]. A Perkin Elmer FT-IR spectrometer (Perkin Elmer Spectrum GX, Shelton, CT, USA) with an attenuated total reflectance (ATR) was used. Ten scans were performed for each sample in a spectral range from 400 to 4000 cm^−1^ with a resolution of 4 cm^−1^.

### 2.8. X-Ray Diffraction Analysis (XRD) and Crystallinity Percentage (CP)

This analysis was carried out according to the methodology described by Remya et al. [15]. An X-ray diffractometer (PANalytical X’Pert PRO) was used. The diffractograms were obtained with a Bragg angle sweep from 5° to 40°, at intervals of 0.02°, operating at 30 kV and 16 mA, with CuKα radiation, wavelength 1.5456 Å, and with a counting time per-step of 100 s. The percentage of crystallinity was calculated using the following Equation (3):(3)$$\%\;Crystallinity=\frac{I_{C}}{I_{a}+I_{c}}\times 100$$
where, Ic: crystallinity area, Ia: amorphous area. 

This analysis was performed in duplicate in each of the treatments

### 2.9. Formulation, Extrusion and Microwave Expansion

The snack formulation was made according to the experimental design of mixtures using a statistical software Desing Expert v.9 (Stat-Ease, Minneapolis, MN, USA) where the concentrations of native starch, modified starch and amaranth flour were established for each treatment in values ranging from 0 to 100%. The extrusion was performed in a single screw extruder (C.W. Brabender Instruments, Inc., Duisburg, Germany) with a 1:1 compression ratio, screw speed of 50 rpm and an exit die with a diameter of 6.5 mm. The temperatures used were 60, 80, 90, and 140 °C in zones 1, 2, 3 and 4 respectively. The extrudate was dried in an oven (Ecoshel FCD 3000, Mc Allen, TX, USA) with air circulation at 40 °C until a humidity of 8% was reached, and later expanded in a microwave oven (LG, Korea) for a time of 2.5 min and power of (900 W) in order to obtain greater expansion and improve the textural properties of the product.

### 2.10. Hardness of the Snack

The hardness of the snack was carried out based on the methodology reported by Limón-Valenzuela et al. [16]. The textural quality of the extrudates was analyzed to determine the hardness and crispness through the application of a texture analyzer TAX-T2 (Stable Microsystems, Surrey, UK) with a cutting blade device. The test condition was 1.0 mm/s pre-test speed, test speed of 2.0 mm/s, 10.0 mm/s post-test speed, and distance of 5 mm. The extrudates were cut to a length of 4 cm and placed horizontally in the texturometer. The force (N) v/s time (s) curves were obtained, where the maximum peak represents the hardness, and the crispness is determined as the linear distance. This analysis was performed three times in each treatment.

### 2.11. Expansion Index of the Snack (EI)

The EI analysis was determined according to the method reported by Gujska and Khan [17], where the diameter of each extruded sample was divided by the orifice diameter of the extruder matrix. Each value was obtained as an average of 10 determinations.

### 2.12. Apparent Density of the Snacks (DAP)

The bulk density of the extrudates was determined according to the technique reported by Pérez-Navarrete et al. [18]. Ten extruded samples of approximately 4 cm were randomly selected, and diameter (d) and length (L) were measured using a digital Vernier (Mitutoyo, Japan). For each sample, five measurements were taken and the average value was calculated. Subsequently, each extrudate was weighed (W_m_) to finally determine the density using Equation (4):(4)$$DAP=\frac{W_{m}}{\pi *{\left(\frac{d}{2}\right)}^{2}*L}$$

### 2.13. Determination of Proteins

Protein determination was performed by method 923.04 [19].

### 2.14. Color Change (ΔE)

This analysis was performed based on the ASTM D6290 standard [20]. The color change was determined for each of the formulations using a colorimeter (Konica Minolta^®^ CR-400/410, Tokyo, Japan), calibrated with a standard (X = 94.9, y = 0.3124, x = 0.3185). The sample was previously grounded and placed in a Petri box to the edge for the measurement; this was performed in the dark to avoid interference; this was done five times for each sample.

### 2.15. Scanning Electron Microscopy (SEM)

The extruded samples were observed in SEM based on the methodology described by Hu et al. [21]. When extruded, a cross section was made and the humidity was adjusted to 2% (*w*/*v*), the samples were placed in a scanning electron microscope (Hitachi SU3500 Scanning Electron Microscope, Tokyo, Japan), to a field of 15 kV to obtain micrographs at 20X, 100X and 500X.

### 2.16. Determination of Resistant Starch (RS)

For the determination of RS, the methodology described by Goñi et al. [22] was used. One-hundred milligrams of the sample was weighed in a centrifuge tube, and 10 mL of KCl-HCl buffer with a pH = 1.5 was added, then 200 μL of pepsin solution was added (the pepsin solution was prepared in a ratio of 25 mg of pepsin with 250 μL of KCl-HCl buffer per sample). The sample was mixed and left in water at 40 °C for 60 min with constant agitation. After this time, the samples were left at room temperature. Later, 9 mL of Tris-maleate buffer were added to each tube at pH = 6.9; 40 mg of α-amylase were added, mixed and incubated for 16 h in a water bath at 37 °C with constant agitation; then the samples were centrifuged for 15 min at 3000× *g* at a temperature of 4 °C. The supernatant was discarded and washed twice with 10 mL of distilled water, centrifuged again and the supernatant discarded. After that, 3 mL of distilled water and 3 mL of KOH (4 M), prepared that same day, were added and left to rest for 30 min at room temperature with constant agitation. Next, 5.5 mL of HCl (2 M) and 3 mL of sodium acetate buffer (0.4 M, pH = 4.75) were added in the same way. After that, 25 μL of amyloglucosidase were added, and left for 45 min in 60 °C water bath, with constant agitation. The samples were centrifuged for 15 min at 3000× *g* at 4 °C and the supernatant was collected and taken to a 50 mL graduated flask. The residue was washed two more times with 10 mL of distilled water each time and the supernatant was combined with the one obtained previously. Then, 50 mL was adjusted, and 50 μL of the sample were taken to determine the glucose released by enzymatic digestion, employing the GOD/PAD method (Glucose Oxidase/Peroxidase), reading the optical densities of the samples at 510 nm.

### 2.17. Experimental Design and Results Analysis

A mixture design with centroid was used in order to evaluate the effect of the composition of the flours in the mixture. Three factors were analyzed: Native Starch (NS), Modified Starch (MS) and Amaranth Flour (AF), where the levels used (0, 16.6, 33.3, 50, 66.6, and 100%) were defined preliminarily by the package Statistical Design Expert v.9.0.3. [23] (Table 1), the maximum values of each ingredient to define the process conditions and treatment behavior in the extrusion equipment were evaluated. The optimization was done in graphic mode based on the values obtained from the characterization of extruded amaranth and chia snacks obtained in local markets of Chihuahua. By superimposing the contour graphs of each response, the area was selected, as well as the optimized values taking the central point of the selected region. The results of the characterization of the raw materials were analyzed using the one-way analysis of variance and the Tukey mean difference test (*p* ≤ 0.05).

## 3. Results and Discussion

### 3.1. Degree of Acetylation of Modified Starch

During the modification, a percentage of acetylation (% Ac) of 1.43 ± 0.11 was achieved, representing a degree of acetylation of 0.05 ± 0.002, classified as low grade. For the use of modified starches in food, the FDA approves GAc in a range of 0.01–0.2, the value obtained is within the specifications [24,25], which makes it a safe ingredient to be included in food for human consumption. The values found are similar to those reported by Shon and Yoo [26] during the acetylation of rice starch in an aqueous medium. On the other hand, they were inferior to those reported by Sodhi and Singh [5] and Colussi et al. [27]; the differences are mainly due to the reaction conditions used.

### 3.2. Thermal Characterization

Table 2 shows the values obtained by DSC of gelatinization onset temperature (T_0_); gelatinization maximum temperature (T_g_); gelatinization final temperature (T_f_); and enthalpy of gelatinization (ΔH_g_) obtained for rice flour (RF), native starch (NS) and modified starch (MS). The starch consists of a mixture of linear (amylose) and branched polymers (amylopectin) which form an ordered granular structure. When the starch is heated in excess of water, the granules undergo an irreversible phase transition known as gelatinization [28].

For the gelatinization onset temperature, the highest value (81.63 °C) was obtained for the rice flour, followed by the native starch (73.15 °C) and finally the modified starch with a value of 61.15 °C. In the case of gelatinization temperatures, the same tendency was observed; the higher value was obtained for rice flour (82.63 °C); for native starch, the value decreased to 78.35 °C while the modified starch decreased to 63.9 °C. In the final temperature of the gelatinization, the rice flour presented a value of 85.25 °C, while the native starch and the modified starch presented temperatures of 83.50 and 66.95 °C respectively, which turned out to be significantly different (*p* < 0.05). For rice flour and native starch, the values were slightly higher than those reported by Singh et al. [4,29] in rice starch and corn respectively, where higher values of onset, peak and final temperature of the gelatinization process are related to the variety of rice, in the same way a lower content of amylose can be because of greater crystallinity in the structure.

There was a significant decrease (*p* < 0.05) in the values of temperature and enthalpies of gelatinization. The extraction process mainly gives the difference between the rice flour and the native starch by using the sodium hydroxide solution in the starch, where there is a decrease in the content of proteins, lipids and fiber, as well as a destabilization in the structure, thus facilitating the entry of water into the granules; this behavior is closely related to the increase in the solubility index. Similar values for native starch and modified starch were reported by Sodhi and Singh [6], who argue that the decrease in gelatinization temperature is given by the introduction of acetyl groups in the molecule. The native starch was subjected to a thermomechanical process causing a pregelatinization of the granules and denaturation of proteins; these phenomena cause the crystalline as well as the amorphous structure to be weakened. So, when the modified starch is heated in the presence of sufficient water, the molecule swells easier and the viscosity increases at lower temperatures than native starch. Nawaz et al. [30] show a similar trend during the acetylation of two rice varieties.

According to Colussi et al. [24], the decrease in the gelatinization temperature is a consequence of the premature opening of the amylopectin double helices and the fusion of starch crystals, induced by acetylation. A tendency of the temperatures to decrease at the beginning, peak and end of gelatinization was observed by these researchers in acetylated rice starch with different concentrations of acetic anhydride (5, 10 and 20%), which corresponds to the behavior represented in Table 1.

On the other hand, a significant decrease (*p* < 0.05) of enthalpy of gelatinization is observed. This decrease indicates the loss of the order of the double helices, as well as the loss of crystallinity in the starch. The highest value (8.64 J/g) corresponds to the rice flour while the lower value (1.35 J/g) was obtained for the modified starch; this decrement is due to the breakdown of the double helices during the acetylation reaction together with the shear effect in the extrusion. A lower value of enthalpy is a reflection of lower stability during gelatinization. Similar behavior was observed by Liu et al. [31] and Colussi et al. [24] in waxy and high amylose rice starch, respectively.

### 3.3. Pasting Properties (RVA)

Table 3 shows the values of maximum viscosity (μ_max_), minimum viscosity (μ_min_), final viscosity (μ_f_), and viscosity of retrogradation (μ_r_) obtained from the viscoamylogram for rice flour, native starch and modified starch.

The maximum viscosity (μ_max_) is a measure of the degree of swelling of the granules. The drop in viscosity from a value to a minimum value (μ_min_) is the decomposition value (breakdown); it is an index of the stability of the pasta during cooking, while the final viscosity (μ_f_) indicates the stability of the cooked pasta. The retrogradation viscosity (μ_r_) is indicated as the difference between the maximum viscosity and minimum viscosity [32]. As can be seen in Table 3, the highest value of μ_max_ (15.8 Pa*s) is presented by rice flour, while the lowest value of μ_max_ (2.88 Pa*s) was obtained in the modified starch. For the minimum viscosity values, the same behavior was observed; a higher value (8.78 Pa*s) that corresponds to the rice flour and a lower value (2.34 Pa*s) was found in the modified starch. The values of μ_r_ exhibited the same trend, in the same way, the highest value (15.35 Pa*s) was obtained in the rice flour and the lowest value (6.45 Pa*s) was obtained in the modified starch. The rice flour also had a higher value (6.1 Pa*s) of retrogradation viscosity while the native starch and the modified starch presented equal values (3.6 Pa*s), lower than that found in rice flour.

For the parameters of μ_max_, μ_min_ and μ_f_, a tendency to decrease was observed. Significant differences (*p* < 0.05) were found for the values of μ_max_ and μ_min_ in rice flour and starches. For the values of final viscosity and retrogradation viscosity, the differences between the native starch and the modified starch did not turn out to be significant (*p* < 0.05), while for these same parameters the differences between the rice flour and the native starch, as well as the rice flour and the modified starch were found to be significant (*p* < 0.05). The viscosity achieved by a paste depends on the level of gelatinization of the starch granules and the degree of decomposition in the molecular structure. The higher viscosity reached by the rice flour is due to the high content of non-gelatinized starch; likewise, the presence of proteins can increase the water retention capacity. Similar viscoamylograms were obtained by Fitzgerald et al. [33] in the rice flour without eliminating and with the elimination of the proteins; likewise, Souza et al. [34] reported viscoamylograms for raw rice flour with behaviors similar to those shown in Figure 1, concluding that rice flour had greater resistance to shear deformation and therefore to heating and cooling cycles.

The viscosity drop observed in the native starch is due to the degradation during the extraction and milling processes; on the other hand, the decrease in the viscosity of the modified starch is directly related to a depolymerization product of the partial pregelatinization of the starch induced by the extrusion process. Similar behavior was observed by Hagenimana et al. [35] and Martínez et al. [36] in extruded rice flour. Puncha-Arnon and Uttapap [37] also observed a viscosity drop for heat-treated rice starch and flour. The decrease in retrogradation is due to the introduction of the acetyl molecules that interact with the OH groups of the glucose monomer that make up the starch polymer. During the retrogradation, a realignment takes place, mainly of the amylose chains; acetylation causes a decrease in the interactions of the starch polymers and increases the affinity for the water molecules [31], branching in the granules and minimizing the tendency to retrograde [38].

### 3.4. Infrared Transmission Spectroscopy with Fourier Transform (FT-IR)

The structural changes caused by the introduction of acetyl groups in the structure of the starch were verified and confirmed by Fourier transform infrared spectroscopy (FT-IR). A comparison of the spectrum of the native starch with the acetylated starch indicated the introduction of the acetyl group; the spectra obtained for the native starch and the modified starch is shown in Figure 2.

The IR spectra of the native and modified starch have global stretches and similar functional groups. For both starches, the intense band can be observed in the range of 3600–3200 cm^−1^, this occurs due to the stretching vibrations of the O–H bond. On the other hand, the bands corresponding to the range 3000–2800 cm^−1^ coincide with the stretching vibrations of the C–H bond. The band found corresponding to the range 900–1500 cm^−1^ is known as the footprint region because it yields unique patterns of the samples and coincides with the stretching of the C–O [39,40]. Similar spectra for rice starch have been described by Fan et al. [41].

For the modified starch, an increase in the absorption band was observed at 1735 cm^−1^; this is the consequence of the stretching of the carbonyl ester, which indicated that acetylation of the starch was done. On the other hand, the absence of a peak in the area 1850–1760 cm^−1^ was observed, this means that the starch was free of unreacted acetic anhydride, and the absence of absorption in the area of 1700 cm^−1^ for the carboxyl group indicated that the product was also free of acetic acid byproduct [24].

Similar spectra with an absorption peak at 1750 cm^−1^ were reported by Colussi et al. [27] during the acetylation of rice starch with a high, medium and low degree of substitution, however, the signal was increased in their investigation because the degree of minor substitution was 0.24 whereas in the present work it turned out to be much smaller with a value of 0.05. Also, studies conducted by Xu et al. [42] demonstrated that an increase in the degree of substitution causes the increase in signal intensity in the acetylated corn starch. Colussi et al. [24] obtained spectra with similar absorption bands in acetylated rice starch in an aqueous medium with a degree of substitution of 0.05; Sodhi and Singh [6] observed the stretch signal of the introduction of the acetyl group at 1730.8 cm^−1^, reaching a degree of substitution of 0.144, and Tang et al. [43] found an absorption peak at wave number 1729 cm^−1^ for rice starch hydrolyzed with pullulanase and acetylated.

### 3.5. X-ray Diffraction Analysis (XRD) and Crystallinity Percentage (CP)

Figure 3 shows the spectra for the native starch and the modified starch obtained by X-ray diffraction analysis. In cereals, the starches exhibit a typical X-ray pattern of type “A”, in the tubers the pattern is of type “B”, while in legumes the pattern is a mixed type “C”. In starch, amylopectin determines the crystallinity by forming a double-helix structure and amylose defines the amorphous regions [29].

In Figure 3, peaks can be observed for native starch at 14.98°, 17.15°, 18.05°, and 23.07°, while for modified starch the peaks of reflection were observed at 15.01°, 17.10°, 17.92°, and 22.08°. This behavior is typical for X-ray diffraction patterns in the “A” type starches corresponding to cereals, where reflections at 15° and 23° are strong [44].

It can also be observed that both patterns turned out to be similar without great changes in the peaks of reflection; the slightly higher intensity found in the native starch is related to a higher crystallinity reaching a value of 14% while the modified starch showed a lower crystallinity observing a value of 10% [4]. Similar patterns for rice starch were obtained by different authors [4,43,45,46,47].

In the X-ray pattern of the modified starch, the changes turned out to be imperceptible. The degree of substitution achieved during the chemical modification, as well as the thermal treatments determine the structural changes in the starch. A low degree of acetylation (0.05) caused minimal destabilization in the amylopectin chains. Studies conducted by Sha et al. [48] for obtaining resistant starch type 4 (chemically modified starch) from rice starch with 0.59% acetylation, humidity of 20% and applying heat time for 20 min showed diffraction patterns without major changes. According to these researchers, an increase in the percentage of acetylation causes a greater loss of crystallinity due to the damage caused in the intermolecular hydrogen bonds. These results are closely related to those obtained in the present investigation.

### 3.6. Characterization of the Snack

The response variables (Hardness, IEE, IEM, DAP and Protein) obtained according to the experimental design are shown in Table 4.

#### 3.6.1. Hardness

Figure 4 shows the response surface for the variable hardness in the final product, observing a linear trend. A first-order model with the representativeness of 72% was obtained (R^2^ = 0.72); however, the model presented a lack of adjustment.

The maximum hardness value (121 N) corresponds to a 100% concentration of modified starch (MS), while the lowest value (32 N) was obtained in the formulation composed of 66.7% of native starch (NS) and 16.7% modified starch (MS) and amaranth flour (AF). A strong tendency to decrease is observed when the AF content in the mixture is increased.

These values are closely related to those reported by Meza et al. [49] where hardness values of 125 N were obtained in snacks with 25% amaranth in its formulation obtained by extrusion at 130 °C in the last zone.

It is known that extrusion is a process that leads to starch gelatinization, protein denaturation and formation of starch-lipid and protein-lipid complexes. The formation of these complexes results in a reduction of the expansion and therefore, an increase in hardness.

The hardness values of the amaranth formulations were higher than those reported by Dokić et al. [50] in corn snacks (18.4–19.2 N); these differences are mainly due to the moisture content in the feed that was lower (16%), and higher temperature (160 °C) in the extrusion die. However, the trend turned out to be similar to that observed in the present investigation; the increase in amaranth content reduced the force required to break the extrudate.

The addition of amaranth in the formulations could lead to changes in the structure of the starches, reducing the hardness compared to the modified starch, the presence of fiber limits the interactions between the polysaccharides and the proteins which could lead to the reduction of the hardness, behavior similar to that explained by Diaz et al. [51], where an extrude based on maize enriched with amaranth, kañiwa and quinoa was evaluated and showed greater hardness in the extruded pure corn (89 N) than the formulations containing the rest of the materials. On the one hand, the plasticizing effect of water can reduce attractive forces between polymers (polysaccharides) by minimizing hardness, while complexation leads to the greater rigidity of polymer structures; the two explanations are consistent since physical characteristics such as hardness depend on the interaction of biocomponents (starch and protein).

#### 3.6.2. Expansion Index (EI).

The response surface graphic (Figure 5A) corresponds to the expansion index of the extrudates (IEE) while Figure 5B represents the response surface for the EI of the snack once expanded into microwaves (IEM).

The IEE response variable was represented with a quadratic model of R^2^ = 0.83 (83% representativeness of the model) and without lack of adjustment. A significant effect (*p* < 0.05) of the linear components was observed, as well as of the quadratic components (NS * AF, MS * AF). The linear terms have a positive effect on the response; the quadratic term NS * AF had a negative impact; while the term MS * AF had a positive impact on the expansion index. The maximum value observed for IEE was 1.41 and corresponded to the native starch (100%). The lowest value for this response was 1.08 obtained in the modified starch extrudate (100%). A trend to decrease is evidenced when the modified starch content in the mixture is increased.

For the response variable IEM, a quadratic model was obtained with a R^2^ = 0.73 (73% representativeness of the model) and without lack of adjustment, identifying a saddle behavior. A negative impact of the quadratic components (NS * AF, MS * AF) on the expansion index was observed while the linear components did not exert significant effects on the response. Although the effect of a microwave expansion did not show higher expansion rates in all the formulations, a higher expansion rate (1.57) was obtained in the modified starch (100%), corroborating the efficiency of the modification processes (acetylation and extrusion). The low values of the expansion index found for the modified starch were caused by changes in its structure; the introduction of the acetyl molecule favors the destabilization in the morphology of the starch granules, causes an increase in the absorption of water, and a decrease in the temperature of gelatinization. However, the temperature in zone 4 of the extruder (140 °C) was not sufficient to achieve rapid evaporation of the absorbed water, limiting the production of air cells and giving rise to a rigid structure; this coincides with the high values of hardness obtained for the modified starch. On the other hand, the humidity (18%) could act with a plasticizing effect; high humidity values decrease the viscosity of the mixture in the extruder and favors the transport of the material minimizing the pressure and therefore the expansion at the output of the equipment [52]. On the other hand, once the extrudate was subjected to microwave expansion, an increase in the expansion index was observed. Similar values of IEE were found for binary mixtures of modified starch-amaranth flour (1.29) and native starch-amaranth flour (1.27).

In the formulations with high levels of amaranth flour, the expansion was limited mainly by the fat content. Percentages of fats exceeding 6% work as lubricants during the extrusion process, cause an irregular flow, decrease the dissipation of mechanical energy (minimizes the action of shear), and exert a protective effect on the dispersion and gelatinization of starch granules, with a high tendency to the formation of amylose-lipid complexes [53,54]. On the other hand, the configuration of the screw extruder is a key point to achieve a greater expansion in products with high contents of free fat; a configuration of the screw that generates a high intensity of shear is needed, as well as a high residence time that allows the absorption of fat. This behavior has been explained by Ilo et al. [55] during the extrusion of a mixture of rice flour and amaranth grains, where they reported a decrease in the rate of expansion by increasing the content of amaranth; this affected the gelatinization rate and the rheological properties of the mixture; they also explained that due to the low amylose content of amaranth (5–7%), a contraction of the air bubbles takes place. The expansion was also limited by the percentage of proteins in the formulations. Proteins exert diluting effects caused by their ability to affect the distribution of water molecules in the matrix, in addition to creating through covalent bonds and electrostatic interactions originated during extrusion [56]. Chávez-Jáuregui et al. [57] reported values of expansion index of 2.32 and 2.84 in extruded amaranth; these values turn out to be greater than those obtained in the present investigation; the differences could be given by the temperatures used that were higher and lower moisture content. Similarly, the use of a die with a smaller diameter (3 mm) produced an increase in pressure immediately before leaving the extruder, favoring the expansion of the product.

According to Souza et al. [34], for further expansion, there are limit values in the composition of the flours to extrude which must not be exceeded. One percent fat, up to 10% protein and 1.5% fiber are recommended. Amaranth flour exceeds these recommended values impacting directly in the expansion. Brennan et al. [53] explained that the fiber alters the amount of water in the extrudate inside the equipment, which can result in an increase in the pressure; however, at the extruder exit, the vaporization does not occur completely due to the availability of the water molecules, producing, as a result, a low expansion. A reduction in the expansion index was observed by Nascimento et al. [58] in the obtention of expanded rice-based crumbs enriched with spent grain from the brewing industry; in this case, nucleation was limited by the fiber content in the formulations affecting the formation processes and the collapse of bubbles in the product.

#### 3.6.3. Apparent Density (DAP).

Figure 6 shows the response surface for the apparent density of the extrudates, with a maximum value of 0.66 g/cm^3^ being observed for the mixtures of the three components. This variable was represented by a special quartic model with a value of R^2^ = 0.98 (98% representativeness of the model) and without lack of adjustment. All the terms of the model were significant, which implies a marked influence of the three components on the density of the extrudate. The lowest apparent density values were obtained for the modified starch (0.48 g/cm^3^, 100%) and the amaranth flour (0.49 g/cm^3^, 100%).

A tendency to increase the density was observed for the mixtures of the three components, in the same way for the mixtures of native starch and modified starch the density showed an increase. The apparent density of the extrudates is inversely related to the expansion index. The density was higher than that reported by Ilo et al. [55] where the values ranged from 0.099–0.226 g/cm^3^. These researchers reported a marked influence of amaranth content on the density; a minimum value was found for 30% of amaranth content; in this same study, a high-density dependence with temperature and humidity during the process of extrusion is also indicated.

The differences can be given by the extension of the gelatinization and the formation of air cells; the use of temperatures in the last zone of the extruder that varied from 150–190 °C and caused a decrease in the viscosity of the mixture which favored the growth of the bubbles, with a subsequent collapse and increase of expansion rates with a consequent decrease in density. Meza et al. [49] reported that increasing the temperature in the extrusion process during the production of amaranth snacks, a lower hardness was obtained due to the increment of porosity in the product. 

On the other hand, the moisture content in the feed zone influences the elasticity of the sample inside the equipment. At lower humidity, the shear rates effect and the degree of gelatinization increase, favoring a higher pressure inside the extruder. On the contrary, moisture values higher than (11–16%), used by Ilo et al. [55], could have a negative effect on the expansion. 

At lower humidity, it increases the shear rates effect and the degree of gelatinization, facilitating an increase in the pressure inside the extruder, a higher moisture value (18%) with respect to that used by Ilo et al. [55] (11–12) in his study could have a negative effect on the expansion. Also, fiber is capable of capturing water molecules with greater force than proteins and starch, inhibiting the release of this in the mixture, which increases density [59].

The apparent density values also turned out to be superior to those obtained by Nascimento et al. [58] in the production of broken rice additives added with co-products from the brewing industry, with reported densities that varied between 0.1681-0.2596 g/cm^3^. However, they agreed that the introduction of materials rich in proteins and fiber increase the density of the expanded ones, behavior similar to that obtained in the present investigation.

#### 3.6.4. Protein

Figure 7 shows the response surface for the protein variable, observing an ascending behavior with an increase of the content of amaranth flour in the formulations. A special quartic model with a value of R^2^ = 0.99 was obtained (99% representativeness of the model) and without lack of adjustment. The highest protein values were represented as expected by the amaranth extrudate (100%); a tendency to increase was observed when increasing its percentage in the formulation. Meza et al. [49] reported a protein content of 6.72 and 10.27% in snacks obtained by extrusion where 25 and 50% of amaranth were incorporated respectively; these values are closely related to those obtained in the present study, where with a lower concentration of amaranth in the formulation, a lower protein content was obtained.

#### 3.6.5. Color Change (ΔE)

Figure 8 shows the response surface for the color change, observing a linear trend. This variable was represented by a first order model with a value of R^2^ = 0.74 (74% representativeness of the model) and without lack of adjustment. Greater values of color change were obtained for the formulations with a higher content of modified starch and amaranth flour. This behavior turned out to be different from that reported by Ilo et al. [55], where they observed a decrease in color variation with the increase of amaranth seeds in the mixture; according to these researchers, high amaranth content minimized the temperature profile and specific mechanical energy, which reduced the development of darkening of the enzymes present in the product. The differences are mainly due to the characteristics of the raw material used; the flour used in the present study was obtained from already processed amaranth (amaranth burst), which favored the development of the Maillard reaction, obtaining higher values of the change of color by increasing the percentage of amaranth.

#### 3.6.6. Crispness

Figure 9 shows the response surface for the crushability of the extrudates, observing a saddle behavior. This variable was represented by a quadratic model with a value of R^2^ = 0.99 (99% representativeness of the model) and without lack of adjustment. It was observed that the highest values of crispness were found when only pure components were extruded (100%). A tendency to decrease was evident for the formulations of the three components and for the MS-AF and NS-AF mixtures.

Compact structures with a minimum of pores have less crispness. During the extrusion cooking of starch-based formulations, the main interactions that occur among the macromolecules of the network are electrostatic forces and hydrogen bonds formed between the hydroxyl groups present in the molecular chains. The main interactions between the macromolecules of the supramolecular network are electrostatic and hydrogen bonds between the hydroxyl groups of the polymers. An increase of proteins in the mixture results in the formation of a network of greater rigidity, due to the active participation of amaranth proteins [57]. The stiffness in the structure likely caused a decrease in the crushability in the extrudates, also limiting the expansion with the consequent increase in density.

### 3.7. Optimization of the Final Product

Figure 10 shows the optimized region obtained for the contour plots of the response variables. The optimum region was obtained for a formulation composed of 60% amaranth flour, 26% native starch and 14% modified starch.

#### 3.7.1. Resistant Starch (RS)

The optimal formulation was characterized as having a resistant starch content of 8.20 ± 2.30%. The formation of RS during the different processes is linked to the retrogradation of amylose. Factors such as the length of the polymer chain, the presence of sugars and lipids, as well as the incubation time and temperature affect retrogradation of amylose. The extrusion has been used to produce resistant starch, during this process are carried out the transformation of the structure of starch granules, the formation of amylose-lipid complexes, or of unstable crystalline structures (type E) [60,61]. Arendt and Zannini [62] reported resistant starch values in raw, expanded, roasted, cooked, and extruded amaranth seeds of 0.5, 0.5, 1.36, 0.25, and 0.66% respectively, reporting significant differences between raw amaranth and roasted and cooked amaranth, while the rest of the treatments showed no differences. These values were found to be lower than that found in the optimal formulation evaluated in the present investigation.

Parchure and Kulkarni [63] determined the contents of resistant starch of rice and waxy amaranth starch subjected to different treatments, including boiling, pressure cooking, roasting, extrusion cooking, frying, and drying in drums. For the extrusion, these researchers adjusted to a humidity level of 14% and used a single screw extruder, at a screw speed of 60 rpm and a compression ratio of 4: 1. They found a resistant starch content for the native rice starch of 6.29% while the amaranth starch presented a value of 5.58%. Roasted, pressure-cooked, drum-dried, extruded, cooked, and fried rice starch contained 5.85, 5.4, 4.5, 3.6, 3.37, and 2.7% resistant starch, respectively. For amaranth starch, the corresponding values were 5.4, 3.6, 3.6, 3.38, 2.39 and 3.38% for similar treatments.

#### 3.7.2. Scanning Electron Microscopy (SEM)

Figure 11A shows the micrographs at three different magnifications (20, 100 and 500×) obtained for the extruded pellets of the optimal formulation. As can be seen, the extrudate has a compact structure with a minimum of pores of an irregular size that varied between 0.40–1.75 cm located mainly in the outer layer. On the other hand, in Figure 11B, the micrographs for the pellets expanded in microwaves at the same magnifications are observed. The differences in the structure are remarkable, there was an increase in the number of pores located through all the entire expanded snack; the pores formed were mostly of regular size with values that ranged from 0.275–1.42 cm.

The reduction of the expansion index, the increase in density, and the decrease in crunchiness are directly linked to the porosity of the extrudates; these results were due to presence of amaranth flour in the formulations. The micrographs exhibit the lubricating effect exerted by the oil content in this pseudocereal, limiting the complete gelatinization of the starch and the development of larger pores. Dokić et al. [50] in their research on an extruded corn snack supplemented with amaranth obtained a similar behavior. A 50% increase in amaranth content reduced the size of the air cells when compared to their corn control (100%).

The differences between the values of rice starch and amaranth starch are attributed to differences in amylose contents in each sample. The treatments of roasting, drum drying, extrusion, and frying caused a decrease in the content of resistant starch in comparison with the native starches. According to these researchers, the decrease is produced due to solubilization caused by macromolecular degradation and also possible dextrinization. The differences found between these results and those reported in the present study could be given by the moisture content used (18%) and a lower screw speed (50 rpm), influencing the processes of gelatinization, degradation and retrogradation.

The differences found between the reported values and the one obtained in the present study can be given by several factors, among them, the methods and the processing variables. According to Eroglu and Buyuktuncer [64], the shorter residence time in extrusion cooking may offer fewer opportunities for amylose chains to associate, which causes a reduction in the resistant starch content, while a longer residence time may provide more opportunities for the association of the amylose chain.

## 4. Conclusions

A degree of substitution in the acetylated starch of a value of 0.05 was reached in the range allowed by the FDA: 0.01–0.2 as safe use for food. A band in the FT-IR spectrum was observed at a wavelength of approximately 1740 cm^−1^ corresponding to the introduction of the acetyl groups in the starch. A significant decrease in the gelatinization temperature and viscosity peaks of the native starch was also observed, confirming that a chemical and physical-mechanical modification was carried out. The expansion was limited by the content of fats in the amaranth, exerting a strong lubricating effect, due to the fiber content which limited the development of vapor bubbles and subsequent collapse, decreasing the number of air cells, as well as their size. An increase in the expansion index was obtained at higher contents of native starch and amaranth, 1.39 and 1.36, respectively, during the direct expansion, while during the microwave expansion, greater expansion was observed at higher content of modified starch with a value of 1.57. The density values were found to be high compared to those reported by the literature; the lowest values were found in mixtures containing only native starch and amaranth (0.57 g/cm^3^). Through the optimization technique, the concentrations of native starch (26%), modified starch (14%) and amaranth flour (60%) with high protein content (10.06%) were determined in comparison to commercial snacks, which represents a high potential to be developed as healthy snacks. The scanning electron micrographs of the optimal formulation showed differences in size, amount and distribution of pores. Resistant starch content of 8.20% was determined, which is higher than the expected value.

## Figures and Tables

**Figure 1 molecules-24-02430-f001:**
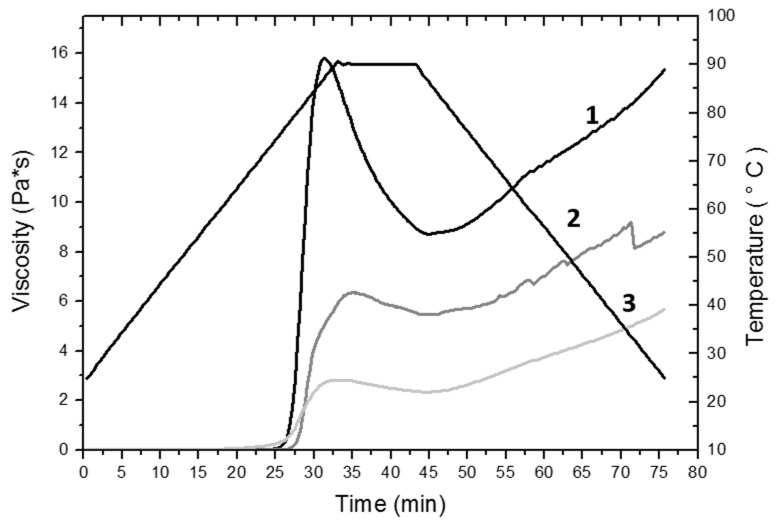
Viscoamylograms obtained for rice flour (1), native starch (2) and modified starch (3).

**Figure 2 molecules-24-02430-f002:**
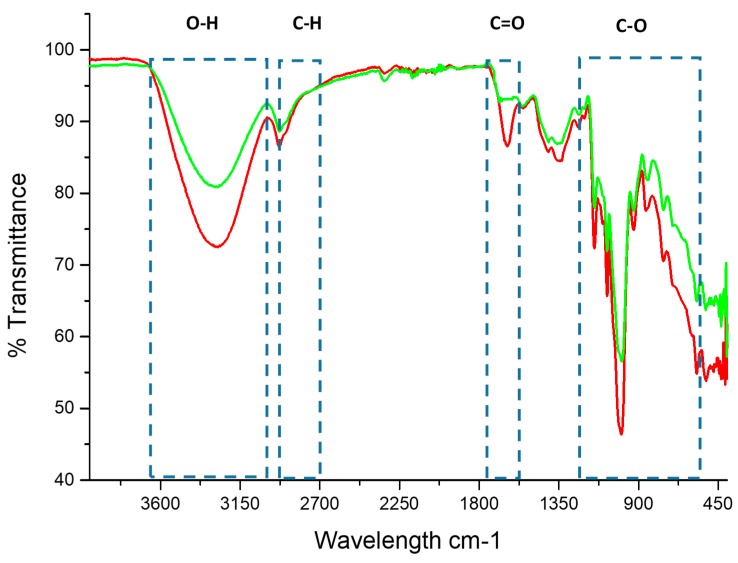
FT-IR spectra of native starch (green curve) and modified starch (red curve).

**Figure 3 molecules-24-02430-f003:**
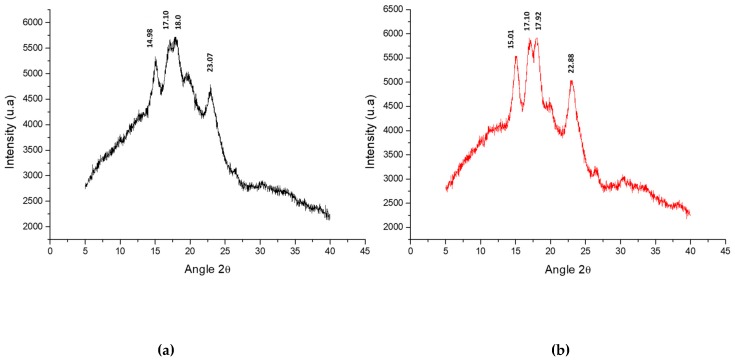
X-ray diffractograms, (**a**) native rice starch (red curve) and (**b**) modified rice starch (black curve).

**Figure 4 molecules-24-02430-f004:**
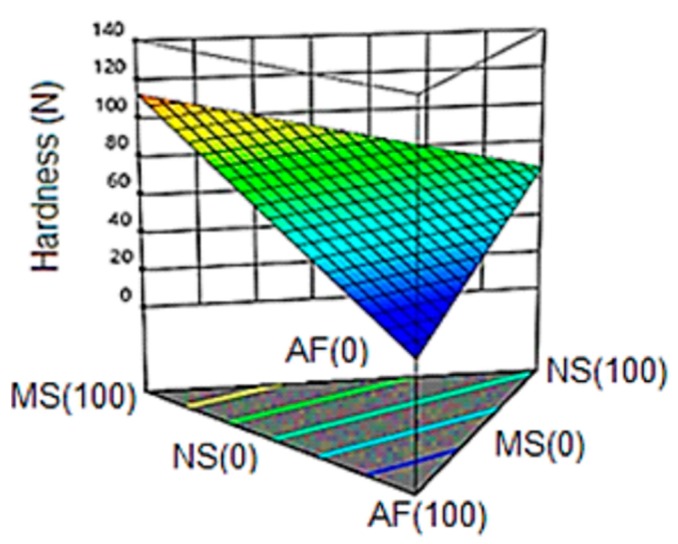
Response surface for the variable hardness in the snack. NS: Native Starch; MS: Modified Starch: AF: Amaranth Flour.

**Figure 5 molecules-24-02430-f005:**
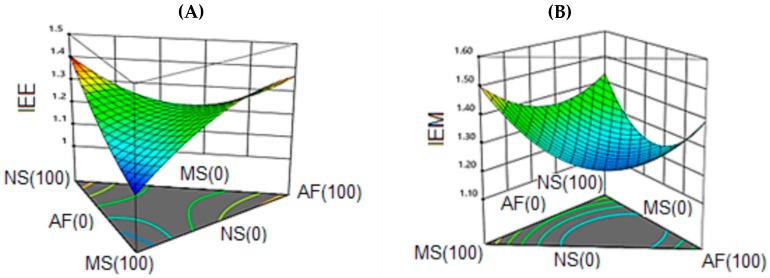
Response surface for expansion index in the snack. **A**) Expansion by Extrusion (IEE), **B**) Microwave expansión (IEM). NS: Native Starch; MS: Modified Starch; AF: Amaranth flour.

**Figure 6 molecules-24-02430-f006:**
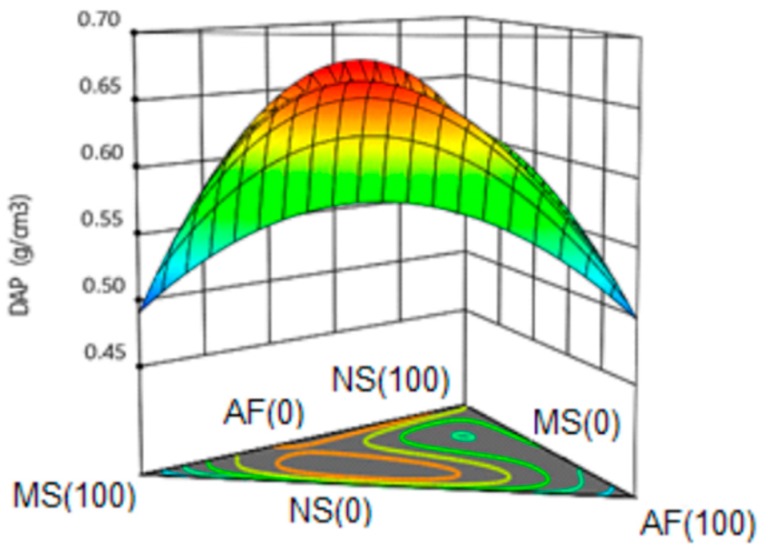
Response surface for apparent density in the snack. NS: Native Starch; MS: Modified Starch; AF: Amaranth flour.

**Figure 7 molecules-24-02430-f007:**
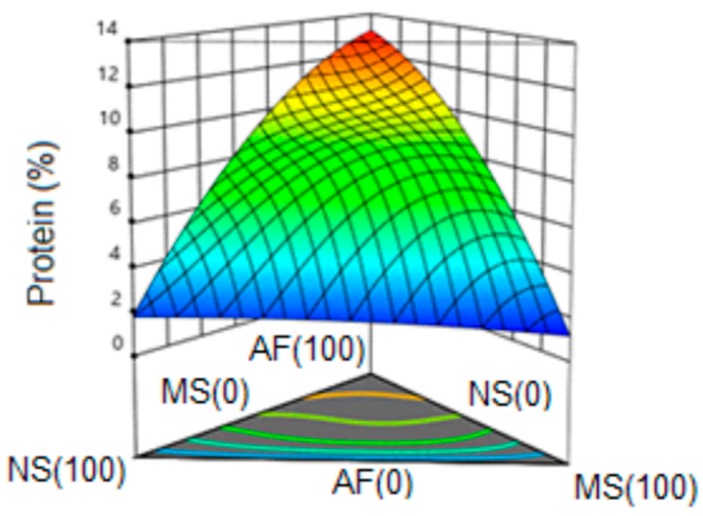
Response surface for protein content in the snack. NS: Native Starch; MS: Modified Starch; AF: Amaranth flour.

**Figure 8 molecules-24-02430-f008:**
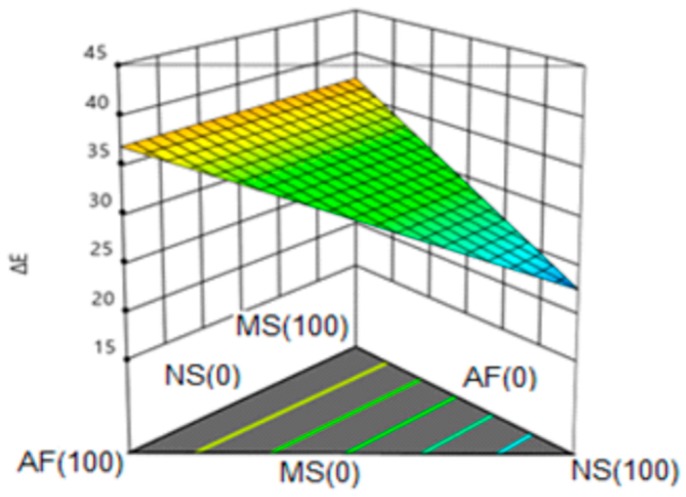
Response surface for total color change (ΔE) in the snack. NS: Native Starch; MS: Modified Starch; AF: Amaranth flour.

**Figure 9 molecules-24-02430-f009:**
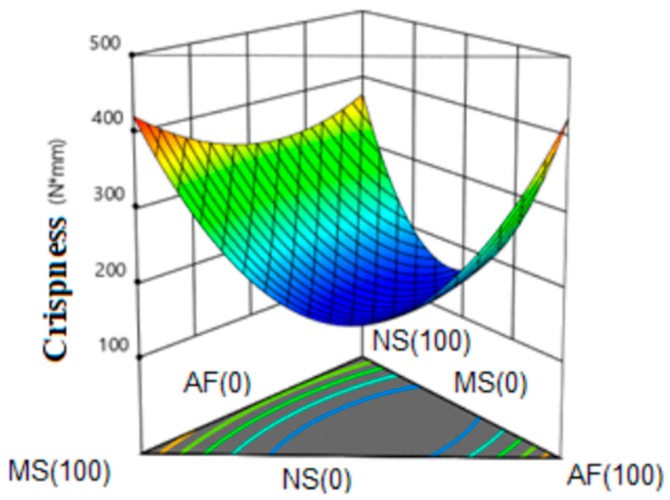
Response surface for crispness in the snack. NS: Native Starch; MS: Modified Starch; AF: Amaranth flour.

**Figure 10 molecules-24-02430-f010:**
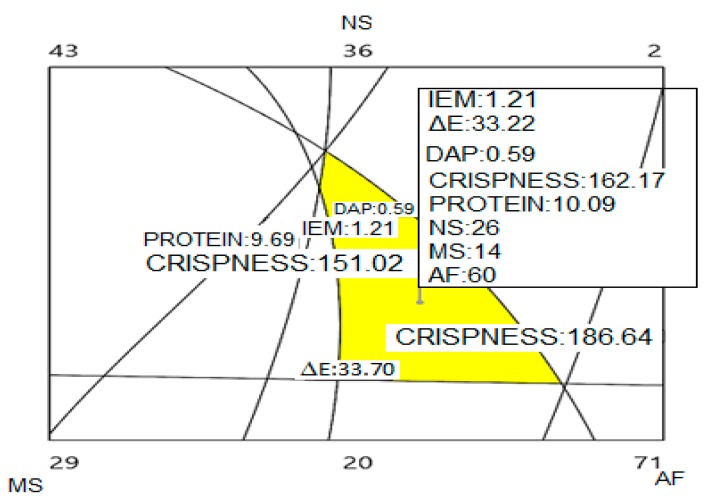
Optimized region obtained from the contour graphs of the characterization of the snack. Microwave expansion index (IEM), Color change (ΔE), Apparent density (DAP) and Crispness (CRI). AN: Native Starch; AM: Modified Starch; HA: Amaranth flour.

**Figure 11 molecules-24-02430-f011:**
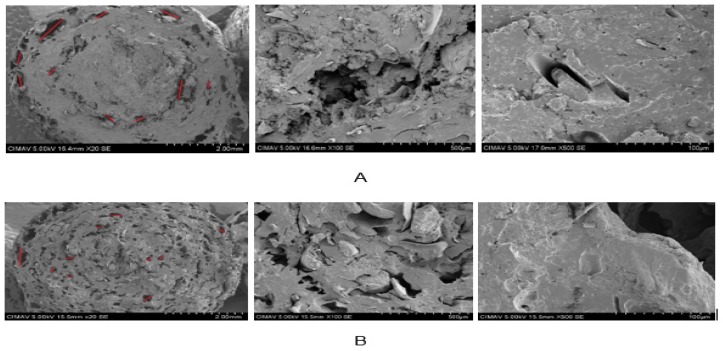
Scanning electron micrographs of the optimal formulation, magnifications 20×, 100× and 500×: **A** (extruded), **B** (expanded).

**Table 1 molecules-24-02430-t001:** Experiment design of mixtures.

Formulations	NS	MS	AF
1	66.67	16.67	16.67
2	50	0	50
3	0	100	0
4	100	0	0
5	33.33	33.33	33.33
6	16.67	66.67	16.67
7	100	0	0
8	0	0	100
9	50	50	0
10	0	50	50
11	50	50	0.00
12	0	100	0
13	0	0	100
14	50	0	50
15	16.67	16.67	66.67

MS: Modified starch, NS: Native starch, AF: amaranth flour.

**Table 2 molecules-24-02430-t002:** Transition temperatures, T_0_, T_g_, and T_f_ and enthalpy of gelatinization (ΔHg, J/g) in rice samples (flour, native starch and modified starch).

	RF	NS	MS
T_0_ (°C)	81.63 ± 0.2^a^	73.15 ± 0.5^b^	61.15 ± 1.3^c^
T_g_ (°C)	82.63 ± 0.2^a^	78.35 ± 0.2^b^	63.90 ± 1.1^c^
T_f_ (°C)	85.25 ± 0.5^a^	83.50 ± 0.1^a^	66.95 ± 1.6^b^
ΔH_g_ (J/g)	8.64 ± 0.0^a^	6.25 ± 0.60^b^	1.35 ± 0.02^c^

* Mean ± standard deviation (n = 3), Tukey test, values that do not share letters for the same row are significantly different *p* < 0.05. RF: rice flour, NS: native starch, MS: modified starch. Transition temperatures, T_0_ onset temperature, T_g_: gelatinization temperature, T_f_: final temperature.

**Table 3 molecules-24-02430-t003:** Viscosity peaks obtained for rice flour, native starch and modified starch at a constant shear rate of 210 s^−1^.

	Rice Flour	Native Starch	Modified Starch
µ_max_, Pa*s, (cP) Heating (25–90 °C, 2 °C/min)	15.8 ± 0.14^a^ (15 800 ± 0.14^a^)	6.35 ± 0.16^b^ (6 350 ± 0.16^b^)	2.88 ± 1.15^c^ (2 880 ± 1.15^c^)
µ_min_, Pa*s (cP) Stability (90 °C, 10 min)	8.78 ± 0.19^a^ (8 780 ± 0.19^a^)	5.50 ± 0.26^b^ (5 500 ± 0.26^b^)	2.34 ± 0.74^c^ (2 340 ± 0.74^c^)
µ_f_, Pa*s (cP) Cooling (90–25 °C, 2 °C/min)	15.35 ± 0.21^a^ (15 350 ± 0.21^a^)	8.77 ± 0.27^b^ (8 770 ± 0.27^b^)	6.45 ± 1.11^b^ (6 450 ± 1.11^b^)
µ_r_, Pa*s (cP)	6.1 ± 0.07^a^ (6 100 ± 0.07^a^)	3.6 ± 0.05^b^ (3 600 ± 0.05^b^)	3.6 ± 0.04^b^ (3 600 ± 0.04^b^)

* Mean ± standard deviation (n = 3), Tukey test, values that do not share letters for the same row are significantly different *p* < 0.05. Values in centipoise are in parentheses.

**Table 4 molecules-24-02430-t004:** Hardness, expansion index, density, and protein content of the snack.

Formulation	NS	MS	AF	Hardness (N)	IEE	IEM	DAP (g/cm^3^)	Protein (%)
1	0.667	0.167	0.167	32	1.32	1.32	0.57	6.07
2	0.500	0.000	0.500	35	1.13	1.15	0.58	9.30
3	0.000	1.000	0.000	121	1.08	1.57	0.48	1.05
4	1.000	0.000	0.000	76	1.39	1.40	0.60	1.72
5	0.333	0.333	0.333	58	1.24	1.24	0.61	8.58
6	0.167	0.667	0.167	54	1.13	1.14	0.65	6.29
7	1.000	0.000	0.000	74	1.41	1.45	0.61	1.83
8	0.000	0.000	1.000	33	1.36	1.37	0.50	13.34
9	0.500	0.500	0.000	112	1.17	1.39	0.67	1.60
10	0.000	0.500	0.500	52	1.29	1.23	0.58	9.11
11	0.500	0.500	0.000	113	1.21	1.38	0.66	1.79
12	0.000	1.000	0.000	120	1.09	1.48	0.50	1.24
13	0.000	0.000	1.000	36	1.37	1.37	0.49	12.99
14	0.500	0.000	0.500	36	1.27	1.28	0.57	9.42
15	0.167	0.167	0.667	37	1.29	1.31	0.61	10.72

NS: Native Starch; MS: Modified Starch; AF: Amaranth flour; IEE: Expansion Index in Extruder; IEM: Expansion Index by Microwave; DAP (g/cm^3^): Apparent Density.

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
