# Peer review of "Development of an Expanded Snack of Rice Starch Enriched with Amaranth by Extrusion Process"

_molecules, 2019, doi:10.3390/molecules24132430_

Round 1

Reviewer 1 Report

Generally the manuscript is readable and the obtained results presented and discussed properly. However before publication the manuscript needs some corrections and supplementations.

Detailed recommendation:

Abstract section:

For the abstract, please add more numerical data, not just trends of changes

Introduction section

Page 2, line 57 – please add more information about acetylation process,

Page 2, line 66 – why you choose amaranth? Could you explain that?

Experimental section:

Page 2, line 71-72 – add more information about used rice and amaranth, what was dry mass of these seeds?

Page 74-75 – what was the yield of meel rice and amaranth?

Page 3, line 105 - in what repetitions was appointed degree of substitution of starch?

Page 4, line 121 - in how many repetitions the determination was made?

Page 4, line 137-145 - in how many repetitions the determinations were made?

Page 5, line 165 – add more information about expansion index of the snack

Results and discussions section

Page 6, lines 245-250 – this information to introduction sections

Table 3: could you add pasting parameters in RVA units?

Table 4: under the table, explain the parameter shortcuts

Author Response

Response to Reviewer 1 Comments

REV 1

Detailed recommendation:

Abstract section:

For the abstract, please add more numerical data, not just trends of changes The changes were realized

Introduction section

Page 2, line 57 – please add more information about acetylation process,The changes were realized

Page 2, line 66 – why you choose amaranth? Could you explain that? Amaranth was selected for its high protein content (18%), as well as the presence of up to three times more essential amino acid content (lysine and tryptophan) compared to other cereals. Reasons why it was considered an important ingredient to fortify the snack obtained.

Experimental section:

Page 2, line 71-72 – add more information about used rice and amaranth, what was dry mass of these seeds? The rice was purchased directly from the supplier without a pericarp, lemma and palea. While the amaranth was already purchased puffed, it was not worked from the plant. In L80 it was specified that a dry grind was used.

Page 74-75 – what was the yield of meel rice and amaranth? The yield was around 90% in rice and amaranth.Sentence was include in text. L81.

Page 3, line 105 - in what repetitions was appointed degree of substitution of starch? This analysis was performed in triplicate in each of the treatments. Sentence was include in text.L117.

Page 4, line 121 - in how many repetitions the determination was made? This analysis was performed in triplicate in each of the treatments. Sentence was include in text.L128.

Page 4, line 137-145 - in how many repetitions the determinations were made? This analysis was performed in duplicate in each of the treatments. Sentence was include in text.L155.

Page 5, line 165 – add more information about expansion index of the snack. The wording was changed for a better understandingof the text. L176-177.

Results and discussions section

Page 6, lines 245-250 – this information to introduction sections. The section was moved to introduction. L58-64.

Table 3: could you add pasting parameters in RVA units? The RVA units (cP) were added. Table 3.

Table 4: under the table, explain the parameter shortcuts. The acronyms under the table 4 were revised to make clear each one. L426.

Reviewer 2 Report

The modified rice starch used for amaranth enriched snack production was developed. The rice starches were analyzed by substitution degree, differential scanning calorimetry, viscosity, Fourier transform infrared spectroscopy and X-ray diffraction. The mixture design was employed for optimal the physical properties of snack. This manuscript is very interesting and brings relevant contribution to the specific area. Overall, the manuscript is recommended for publication after major revision. The following specific comments may be considered while revising the manuscript.

1.      Line 14, The raw material should be Specified. The raw material (rice flour, native starch, and modified rice starch) was evaluated by the ……

2.      Line 24, the abbreviation must have a full name, when first occurrence in the article. For example, NS, EI, BD, MS, AF, P, RS. Please check the whole manuscript.

3.      Line 66-68, Delete “Providing added value to a by-product of the food industry and offering an alternative of ready-to-eat food with a contribution of nutrients, fiber and low fat, for…….”. The nutrients, fiber and low fat did not analysis in this study. Replace this paragraph with the things done in this study. For example, the mixture design was employed for optimal the physical properties of snack.

4.      Line 76-85, Is it the procedure for preparing rice starch? Please Specify.

5.      Line 86-96, Is it the procedure for preparing modified starch? Please Specify.

6.      Line 146, Combine 2.9 and 2.17 in one section.

7.      Line 209, move this section into 2.9.

8.      Line 208, How to calculate resistant starch from the optical densities of the samples at 510 nm? Please explain in details or give a formula.

9.      Line 266, superscript a, b, c letter.

10.  Line 312, superscript a, b, c letter.

11.  Line 383, superscript -1.

12.  Line 418, say some thing in section 3.6. Don't just put a table there.

13.  Line 604, the hardness, expansion index (EI), apparent density (DAP), protein, color change (ΔE), and crispness are the response. How to determine the optimization of the final product? Please clarified.

14.  Line 613, unit of resistant starch content. 8.20 ± 2.30 “%”.

Author Response

Response to Reviewer 2 Comments

REV 2

1.        Line 14, The raw material should be Specified. The raw material (rice flour, native starch, and modified rice starch) was evaluated by the …… The change was realized

2.        Line 24, the abbreviation must have a full name, when first occurrence in the article. For example, NS, EI, BD, MS, AF, P, RS. Please check the whole manuscript.The change was realized

3.        Line 66-68, Delete “Providing added value to a by-product of the food industry and offering an alternative of ready-to-eat food with a contribution of nutrients, fiber and low fat, for…….”. The nutrients, fiber and low fat did not analysis in this study. Replace this paragraph with the things done in this study. For example, the mixture design was employed for optimal the physical properties of snack. The change was realized

4.      Line 76-85, Is it the procedure for preparing rice starch? Please Specify. Yes it is, this methodology was used in other scientific journal for rice starch isolation  as:

Example: J Food Process Preserv.2018;e13428. https://doi.org/10.1111/jfpp.13428

5.      Line 86-96, Is it the procedure for preparing modified starch? Please Specify. Yes it is, this methodology was used in other scientific journal for modified rice starch isolation  as:

Example: J Food Process Preserv.2018;e13428. https://doi.org/10.1111/jfpp.13428

6.      Line 146, Combine 2.9 and 2.17 in one section. The authors do not consider it convenient to combine these sections, since point 2.17 makes reference to the handling of data of raw material, extruded and optimized product.

7.      Line 209, move this section into 2.9. It was not done since the comment is not clear

8.      Line 208, How to calculate resistant starch from the optical densities of the samples at 510 nm? Please explain in details or give a formula. This analysis was sent to perform in Development Food and Research Center (CIAD, Mexico) for our research..With the following formula AR was determinated:   

%AR= (Glucose (µg/mL)*volume*dilution*100*0.9)/1000*dry sample(mg))

9.      Line 266, superscript a, b, c letter. The change was realized

10.  Line 312, superscript a, b, c letter. The change was realized

11.  Line 383, superscript -1. The change was realized

12.  Line 418, say some thing in section 3.6. Don't just put a table there. The change was realized

13.  Line 604, the hardness, expansion index (EI), apparent density (DAP), protein, color change (ΔE), and crispness are the response. How to determine the optimization of the final product? Please clarified. In section 2.17, L225-228 specify: “The optimization was done in graphic mode based on the values obtained from the characterization of extruded amaranth and chia snacks obtained in local markets of Chihuahua. By superimposing the contour graphs of each responses, the area was selected, as well as the optimized values taking the central point of the selected región”.. using a package Statistical Design Expert v.9.0.3 in optimization section.

14.  Line 613, unit of resistant starch content. 8.20 ± 2.30 “%”. The change was realized. L618.